# External Beam Accelerated Partial Breast Irradiation in Early Breast Cancer and the Risk for Radiogenic Pneumonitis

**DOI:** 10.3390/cancers14143520

**Published:** 2022-07-20

**Authors:** Oliver J. Ott, Wilhelm Stillkrieg, Ulrike Lambrecht, Tim-Oliver Sauer, Claudia Schweizer, Allison Lamrani, Vratislav Strnad, Carolin C. Hack, Matthias W. Beckmann, Michael Uder, Rainer Fietkau, Luitpold Distel

**Affiliations:** 1Department of Radiation Oncology, Universitätsklinikum Erlangen, 91054 Erlangen, Germany; willi.stillkrieg@uk-erlangen.de (W.S.); ulrike.lambrecht@uk-erlangen.de (U.L.); tim-oliver.sauer@uk-erlangen.de (T.-O.S.); claudia.schweizer@uk-erlangen.de (C.S.); allison.lamrani@uk-erlangen.de (A.L.); vratislav.strnad@uk-erlangen.de (V.S.); rainer.fietkau@uk-erlangen.de (R.F.); luitpold.distel@uk-erlangen.de (L.D.); 2Comprehensive Cancer Center Erlangen-EMN, 91054 Erlangen, Germany; 3Department of Gynecology, Universitätsklinikum Erlangen, 91054 Erlangen, Germany; carolin.hack@uk-erlangen.de (C.C.H.); matthias.beckmann@uk-erlangen.de (M.W.B.); 4Department of Radiology, Universitätsklinikum Erlangen, 91054 Erlangen, Germany; michael.uder@uk-erlangen.de; 5Radiobiologic Laboratory, Department of Radiation Oncology, Universitätsklinikum Erlangen, 91054 Erlangen, Germany

**Keywords:** accelerated partial breast irradiation, early breast cancer, radiogenic pneumonitis, lung dose constraints, individual radiosensitivity

## Abstract

**Simple Summary:**

Accelerated partial breast irradiation represents a well-established and safe treatment alternative for selected patients with early stage breast cancer after breast-conserving surgery. In a prospective trial on external beam partial breast irradiation, the risk for radiogenic pneumonitis is analyzed. After a median follow-up of 56 (1–129) months, radiogenic pneumonitis grade 2 appeared in only 1/170 (0.6%) patients. Compared to standard whole breast irradiation, the radiation doses applied to the lungs were very low. The patient with the radiogenic pneumonitis was found to have a generally increased radiation susceptibility. Using accelerated partial breast irradiation, the risk of radiogenic pneumonitis appeared quite low and may be limited to very exceptional cases associated with innate risk factors with an increased radiation susceptibility.

**Abstract:**

In order to evaluate the risk for radiation-associated symptomatic pneumonitis in a prospective external beam accelerated partial breast irradiation (APBI) trial, between 2011 and 2021, 170 patients with early stage breast cancer were enclosed in the trial. Patients were eligible for study participation if they had a histologically confirmed breast cancer or an exclusive ductal carcinoma in situ (DCIS), a tumor size ≤3 cm, free safety margins ≥2 mm, no involved axillary lymph nodes, tumor bed clips, and were ≥50 years old. Patients received APBI with 38 Gy with 10 fractions in 10 consecutive working days. The trial was registered at the German Clinical Trials Registry, DRKS-ID: DRKS00004417. Median follow-up was 56 (1–129) months. Ipsilateral lung MLD, V20, and V30 were 4.3 ± 1.4 Gy, 3.0 ± 2.0%, and 1.0 ± 1.0%, respectively. Radiogenic pneumonitis grade 2 appeared in 1/170 (0.6%) patients two months after radiotherapy. Ipsilateral MLD, V20, and V30 were 6.1 Gy, 7, and 3% in this patient. Additionally, individual radiosensitivity was increased in this specific patient. Compared to WBI, APBI leads to lower lung doses. Using APBI, the risk of symptomatic radiogenic pneumonitis is very low and may be limited, with an ipsilateral V20 < 3% to very exceptional cases associated with innate risk factors with an increased radiation susceptibility.

## 1. Introduction

Accelerated partial breast irradiation (APBI) represents a well-established and safe treatment alternative for selected patients with early stage breast cancer after breast-conserving surgery. This is, among other recommendations, based on the results of the St. Gallen International Breast Cancer Conference [1,2]. With low-risk patient selection provided, several randomized trials compared APBI regimens to standard external beam whole-breast irradiation (WBI) and proved non-inferiority [3]. Today, APBI is the best evaluated research topic in breast radiotherapy. Because of the dramatically reduced treatment volume, patients with APBI usually experience a lower rate of radiotherapy-associated side effects compared to WBI [4,5]. In this analysis, we focused on lung dose and lung toxicity of a phase 2-study cohort prospectively treated within the German External Beam APBI-trial [6]. Additionally, individual radiosensitivity was analyzed in the only case with a symptomatic radiogenic pneumonitis CTCAE v3.0 grade 2.

## 2. Materials and Methods

Between 2011 and 2021, 170 women with pT1-2pN0 breast cancer participated in this phase 2-trial. All studies on human beings reported in this manuscript were conducted in accordance with national law and the Helsinki Declaration of 1975 in its current, revised form, and with the approval of the ethics committee of the Friedrich-Alexander University of Erlangen. Informed written consent declarations were collected from all study participants. The trial was registered at the German Clinical Trials Registry, DRKS-ID: DRKS00004417.

At the beginning, patients obtained closed cavity breast conserving surgery (BCS). The surgical standard procedure for the axilla was a sentinel node biopsy combined with a level I dissection. In case of pure DCIS, axillary surgery for staging purposes was optional. After BCS, usually an interval of 4–6 weeks was dedicated for wound healing and for proper histological workup of the tumor specimen to guarantee proper patient selection. APBI was planned to begin between the 5th and the 10th postoperative week. In case of additional chemotherapy, APBI started within 28 days thereafter.

In accordance with the study plan, patients were suitable for the trial if they had a histologically confirmed invasive breast cancer or pure DCIS (Union for International Cancer Control (UICC) stage 0-IIA), a tumor diameter ≤ 3 cm, complete resection, clear margins ≥ 2 mm (in case of DCIS ≥ 5 mm), at least six negative axillary lymph nodes (pN0), or singular nodal micro-metastasis (pN1mi), or negative sentinel node biopsy (pN0sn), or a clinically negative axilla in case of DCIS (cN0), no distant metastasis or contralateral breast cancer, tumor bed clips, and were ≥ 50 years old. Patients were excluded from study participation if they showed any signs of a multifocal growth pattern in the diagnostic mammogram, evidence of residual micro-calcifications, an extensive intraductal component (EIC), vessel invasion (L1, V1), involved or unknown margins (R1/Rx), or were pregnant.

Computed-tomography (CT)-based APBI planning (see Figure 1) and every single irradiation fraction were applied using a standardized endexspiratory breath hold protocol to minimize intrafractional motion at a dedicated linear accelerator combined with the ExacTrac X-ray patient position monitoring system (Brainlab, Feldkirchen, Germany) using fiducial markers for reproducible daily positioning and to reduce the interfractional positioning error.

Generally, the planning target volume (PTV) enclosed the tumor bed with combined postoperative and radiooncological safety margins. The tumor bed was defined by the use of titanium marker clips, preoperative mammography, ultrasound examinations, and postsurgical planning CT-scans. The clinical target volume (CTV) was expanded around the tumor bed using the postoperative safety margins by adding space to get a combined safety area of 20 mm in all six directions [7]. After expansion, the CTV was limited to the chest wall and the skin excluding 3–5 mm below the surface. The PTV was accomplished by a 15 mm expansion in all six directions excluding the ipsilateral lung. Patients were given a total dose of 38.0 Gy at the ICRU-50 reference point using tangential fields in 3.80 Gy daily fractions, five times a week. The study protocol accepted partial breast irradiation acceleration with 1–2 daily fractions, but at the start of the study the principal investigator decided to apply only one fraction per day to prevent potential toxicity. Sequential chemotherapy and/or hormonal therapy were allowed following the recommendations of the local partners.

A whole blood sample of the patient with symptomatic radiogenic pneumonitis was drawn five years after completion of radiation therapy. Half of the blood samples were irradiated with a single dose of 2 Gy, the other part remained unirradiated. Lymphocytes were stimulated with phytohemagglutinin and were arrested in metaphase by colcemid 48 h later. Chromosomes 1, 2, and 4 were stained with chromosome-specific probes. The chromosomal aberrations were scored as breaks per metaphase (B/M), corresponding to the breaks that were necessary for aberrations to occur. At least 150 metaphases were evaluated. The chromosomal aberrations of the index patient were compared to a cohort of 218 healthy individuals and 274 individuals who were suffering from rectal cancer [8,9]. From a B/M value ≥ 0.55, a significantly increased aberration rate was assumed.

Follow-up was conducted every three months for two years following BCS, every six months in years 3–5, and at 12-months intervals thenceforth. Local control and survival parameters, as well as toxicity and cosmetic outcomes were checked with mammograms at guideline-based regular intervals and clinical examinations. Further imaging, e.g., computed tomography scans, was individually added in case of symptoms.

Early side effects were graded according to the Common Terminology Criteria for Adverse Events v3.0 (CTCAE; 10 June 2003), and late side effects correspondent to RTOG/EORTC criteria [10] and Lent Soma tables [11]. For this analysis, we focused on lung toxicity. Multiple lung dose parameters were analyzed from all 170 study patients and compared to the doses registered in the symptomatic patient with a radiogenic pneumonitis.

Data management and statistics were carried out with IBM SPSS Statistics for MS Windows (SPSS Inc., Chicago, IL, USA), release 24. Primary endpoint of the presented study was local control, but these results were not the subject of this analysis and will be published separately. Because of the symptomatic patient with radiogenic pneumonitis, an exploratory analysis of the applied lung doses was performed in this prospective study cohort and compared to previously published data.

## 3. Results

Between 2011 and 2021, 170 patients were treated within this trial. The median age was 62 years (range, 49–86). All patients received closed-cavity breast-conserving surgery.

The time between BCS and APBI was 62.5 days (range, 10–263) for the whole group, and for women with or without additional chemotherapy, it was 221 days (range, 143–263) and 61 days (range, 10–126), respectively. Radiotherapy was applied within ten consecutive working days (range, 9–15). All 170 patients had fiducial titanium markers at the tumor bed to provide appropriate positioning for every fraction and received the prescribed dose. Nearly all patients (169/170, 99.4%) received planning computed-tomography and each single irradiation fraction with an endexspiratory breath hold technique and eight (range, 4–14) static conformal radiotherapy fields. The first treated patient was treated with a conventional free breathing technique. The mean size of the planning target volume (PTVeval: planning target volume excluding extracorporeal areas, the zone of 3-5 mm below the skin surface, and the thoracic wall) was 178 ± 86 mL.

The majority of the patients (161/170; 94.7%) received adjuvant antihormonal treatment, usually with aromatase inhibitors. Eight patients (11/170; 6.5%) received chemotherapy. Six patients with pure DCIS (3.5%) received no additional drug treatment.

At the time of analysis, the median follow-up was 56 (1–129) months. A detailed analysis of the irradiation doses to both lungs was performed for the complete study collective (see Table 1). The predefined protocol constraints for the ipsilateral (D30 < 15 Gy) and contralateral lungs (D5 < 5.7 Gy) were met for the whole collective. Ipsilateral D30 and contralateral D5 were 5.0 ± 1.9 and 1.7 ± 1.4, respectively. Ipsilateral Dmean, V20, and V30 were 4.3 ± 1.4 Gy, 3.0 ± 2.0 %, and 1.0 ± 1.0 %, respectively.

During follow-up, one patient (1/170; 0.6%) experienced symptomatic radiogenic pneumonitis grade 2. This patient was diagnosed with an invasive breast cancer in the upper outer quadrant of the right breast (TNM: pT1c pN0(sn) M0; estrogen and progesterone receptor expression positive; MIB1 25%; Her2/neu negative) and received breast conserving surgery in December 2015. Prior to the surgery, the non-smoking patient presented no pathological findings regarding her cardiopulmonary status, and the condition of an insulin-dependent diabetes type 2 was known already. Right after surgery, the patient started taking oral aromatase inhibitors, and adjuvant chemotherapy was not given because of the low-risk profile. APBI was applied with eight static fields up to the prescribed dose in February 2016 within 10 consecutive working days (the radiotherapy dose distribution is as shown in Figure 1). After completion of radiotherapy, the patient developed a radiodermatitis CTCAE v3.0 grade 1 in the irradiated skin area of the breast with no other signs of early toxicity. At the first follow-up visit, in March 2016, the radiodermatitis was almost non-existent and no further signs of any toxicity were diagnosed. Eight weeks after completion of radiotherapy the patient reported of a dry cough without fever. A diagnostic computed-tomography confirmed the diagnosis of a typical lung tissue fibrosis after radiogenic pneumonitis in August 2016 (see Figure 2).

At the next follow-up visit in September 2016, the dry cough completely vanished without any specific therapy. The lung doses of the patient with the symptomatic radiogenic pneumonitis did not exceed any study-specific constraints with an ipsilateral D30 of 7.0 Gy and a contralateral D5 of 0.5 Gy. Compared to the complete study collective, 13/20 of the evaluated lung dose parameters of the symptomatic patient exceeded the 97.5 percentile of the dose distribution, especially dose parameters related to the ipsilateral side (see Table 1).

Because of the clinically evident radiotherapy-related pneumonitis without inadequate high doses to the lung tissue, an individual radiation sensitivity analysis was performed for this patient. A detailed medical history of the patient regarding radiosensitivity was taken. No genetic burden was found and no breast cancer gene mutations are known. When questioned about connective tissue diseases, the patient reported a pre-existing vitiligo. Radiosensitivity testing by three color fluorescence in situ hybridization (see Figure 3) revealed a chromosome breakage frequency of 0.79 B/M and thus a clearly increased radiosensitivity above the threshold of 0.55 B/M. Compared to a cohort of healthy individuals and patients with rectal cancer, the index patient is above the 95% percentile of both cohorts. The background value of 0.071 B/M was slightly lower than the average of 0.084 B/M. A laboratory study of standard autoantibodies gave no indication of a specific autoantibody. Similarly, exposure of the patient’s serum to different cell lines gave no specific binding of autoantibodies.

## 4. Discussion

The comparison of lung doses and the determination of reliable dose constraints to avoid radiogenic pneumonitis is a very difficult task, because relevant trials often do not provide comprehensive dosimetric data, due to the use of different radiotherapy techniques (external beam irradiation vs. interstitial brachytherapy and others), target volumes (APBI vs. WBI with and without regional nodes), and fractionation schedules (normo- vs. hypofractionation), and because the comparison of data from prospective clinical trials with in silico simulations has some limitations. It has previously been reported that the reduced target volumes of APBI may lead to lower doses to the related organs-at-risk (OAR) [12,13], which has been updated regarding lung dose constraints in Table 2. This comprehensive analysis of the lung doses revealed comparatively low values in the 170 study patients. The mean ipsilateral V20 was 3.0 ± 2.0 %. In comparison to the representative WBI data (see Table 2), this results in a 3–6-fold dose reduction. Furthermore, the data presented in Table 2 document certain advantages for deep inspiration breath hold (DIBH) techniques compared to the use of free breathing techniques in WBI. The advantages of using APBI techniques appear to be even more pronounced. To date, the question whether different APBI techniques are systematically superior to others regarding the doses to the lungs cannot be answered adequately because of the absence of comparative prospective data. In silico dose distribution simulations are of some interest but did not clearly prove one APBI technique better than another because of their often limited sample sizes and virtual nature. The study protocol-defined OAR-constraints for both lungs were met with ease in all patients in our study (see Table 1).

In a comparative in silico planning study of 32 APBI patients, similar mean ipsilateral lung V5 percentages were described for multicatheter interstitial brachytherapy and stereotactic external beam APBI: 30.6 vs. 30.7%. In our trial, we found a comparable ipsilateral V5 of 30.1% [14]. The average ipsilateral mean lung doses (MLD) were about 5% of the prescribed dose in both brachytherapy and external beam plans. The ipsilateral MLD in our clinical trial was a little higher, with 11 % of the prescribed reference dose. In general, this analysis demonstrated no major dosimetric differences between interstitial brachytherapy and stereotactic external beam APBI. Another in silico dosimetric planning study in 40 planning CTs of APBI patients were performed with conformal and three different intensity modulated APBI techniques, reported on mean ipsilateral lung doses (MLD) between 5.6 and 9.9%, with V10 and V30 ranging from 13–36% and 4–6%, respectively [15]. The ipsilateral MLD, V10, and V30 were comparable with 11%, 11%, and 1% in our trial. For the contralateral lung, D5 and V5 ranged between 1–5% and 0–10%, compared to 4% and 1% in our clinical trial. The authors concluded that the OAR doses of all four external beam APBI techniques were tolerable.

The reports on radiogenic pneumonitis following APBI are quite rare. For the first time, Recht et al. published four cases of symptomatic pneumonitis after external beam 3D-conformal APBI in 2009 [16]. In the prospective dose-escalation trial, external beam 3D-conformal APBI was prescribed in two dose level cohorts with 8 × 4 Gy (n = 99) and 9 × 4 Gy (n = 99). All four patients with symptomatic radiogenic pneumonitis received 36 Gy (4/99, 4%), and an ipsilateral lung V20 > 3% was discussed to be associated with an increased risk of pneumonitis. In our trial, the symptomatic pneumonitis rate was 1/170 (0.6%) and the average ipsilateral lung V20 for all patients was 3.0 ± 2.0%, whereas it was 7.0% in the symptomatic patient. Compared to the whole study cohort, the symptomatic patient demonstrated an evident increased ipsilateral lung V5 (47 vs. 30%), and the majority of the ipsilateral and total lung doses were found beyond the 97.5 percentile values (see Table 1). This means that the symptomatic pneumonitis might be related to the in comparison higher lung dose and, furthermore, that the lung dose constraints as defined in the study protocol were not sufficient to avoid these kinds of side effects. Our data support the recommendation of Recht et al. that the ipsilateral lung V20 should be <3% [16]. A second working group reported on the incidence of radiogenic symptomatic pneumonitis after 3D-conformal external beam APBI.

Fifty-five patients received 30 Gy in five fractions over 10 days. After a median follow-up of 30 (18–46) months, three patients (5%) developed grade 2 symptomatic pneumonitis at five, eight, and nine months after APBI. In this trial, ipsilateral lung V20 (≥3 vs. <3%) and V5 (≥20 vs. <20%) were not associated with the risk of symptomatic radiation pneumonitis [17].

Finally, the reasons contributing to the development of a symptomatic radiogenic pneumonitis remain unclear because of the limited available data. It is possible that all the published cases had an increased susceptibility for radiation effects. Therefore, in our patient, we performed individual radiation sensitivity analysis, as established at our department [8,9]. The index patient had a significantly increased radiation sensitivity compared to a mean value of 0.46 B/M in the control groups (see Figure 3). From a threshold value of 0.55 B/M, we already recommend dose adjustments in prospective studies. Regarding the presented symptomatic patient, we would have assumed an increased effect of radiation of 15–30% with a mean of 22.5%. A single dose of 3.8 Gy used here would have had the same effect as 4.4 to 4.9 Gy. The total dose of 38 Gy would correspond to 44 to 49 Gy. Due to the non-linearity of the radiation effect, the biological effectiveness would have been even higher. However, such an increased radiosensitivity would not necessarily lead to an adverse therapeutic outcome. A therapy with specific fractional doses and total dose is defined as a certain risk per year of suffering an undesirable therapeutic consequence, depending on the radiosensitivity of the exposed individual [18]. Other distress caused to the tissue during or after radiation therapy can again significantly increase the risk of side effects.

The cause of the increased radiosensitivity in the patient is unclear. Radiosensitivity may be due to mutations in genes related to repair, signal transduction, cell cycle regulation, and cell death control. Thus, several hundred genes may be involved. Only the 40 most frequent genes in which there were no specific mutations were analyzed in the patient. It was visible that the patient had vitiligo. In recent years, we have found an increased sensitivity to radiation in several patients with vitiligo (personal information). Vitiligo belongs to the autoimmune diseases, where an acquired increased radiosensitivity has been discussed for a long time [25]. It is quite possible that this patient developed an increased sensitivity to radiation due to her autoimmune disease, which ultimately led to the therapy-related side effects.

## 5. Conclusions

The smaller treatment volumes of APBI lead to lower lung doses compared to WBI. In this prospective external-beam APBI trial, symptomatic radiogenic pneumonitis grade 1 appeared in 0.6% (1/170) of the cases. With an ipsilateral lung V20 <3%, the risk of symptomatic pneumonitis may be limited to very exceptional cases associated with innate risk factors for an increased radiation susceptibility.

## Figures and Tables

**Figure 1 cancers-14-03520-f001:**
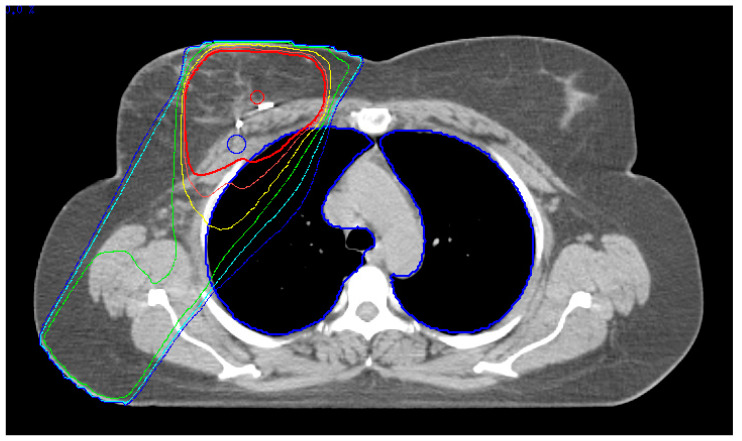
External beam partial breast irradiation treatment plan and dose distribution. APBI of a right-sided breast cancer after closed-cavity breast-conserving surgery with prepectoral tumor bed clips. Isodoses: red bold 95%, red thin 90%, yellow 80%, green 60%, light blue 40%, and dark blue 30%. Total lung organ-at-risk contour in dark blue.

**Figure 2 cancers-14-03520-f002:**
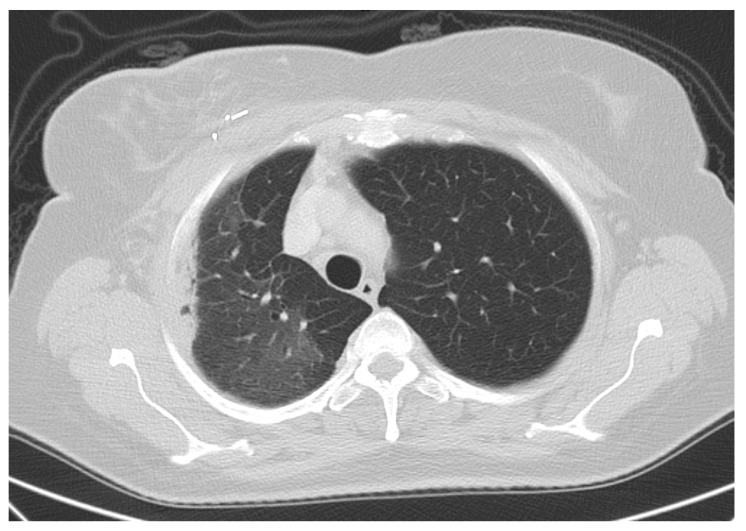
Computertomographic fibrotic signs of radiation-associated pneumonitis six months after completion of radiotherapy. Diagnostic computed tomography of the patient with a typical lung tissue fibrosis after symptomatic pneumonitis six months after completion of APBI. The tumor bed clips indicate the similar position as in Figure 1.

**Figure 3 cancers-14-03520-f003:**
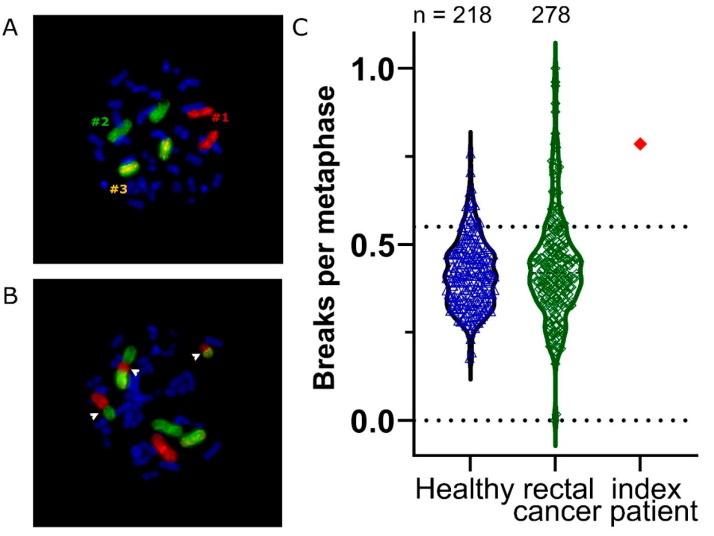
Individual radiation sensitivity analysis by 3-color fluorescence in situ hybridization. (**A**) Metaphase with painted chromosomes #1 in red, #2 in green, and #3 in yellow. (**B**) A metaphase with chromosomal aberrations marked by white arrow heads. (**C**) Breaks per metaphase of the index patient compared to a healthy control cohort and a cohort suffering from rectal cancer.

**Table 1 cancers-14-03520-t001:** Lung dose analysis of all patients in comparison to the patient with symptomatic pneumonitis.

	Trial Results (n = 170) (Mean ± SD)	Lung Dose Constraints (Study Protocol)	Symptomatic Patient (n = 1)
**Ipsilateral lung**			
MLD [Gy]	4.3 ± 1.4	-	6.1 *
D30 [Gy]	5.0 ± 1.9	<15	7.0 *
D2 [Gy]	23.5 ± 7.3	-	32.5 *
D1 [Gy]	28.2 ± 7.2	-	35.0
V5 [%]	30.1 ± 13.1	-	47.0 *
V10 [%]	10.5 ± 5.6	-	15.0
V20 [%]	3.0 ± 2.0	-	7.0 *
V30 [%]	1.0 ± 1.0	-	3.0 *
**Contralateral lung**	0.6 ± 0.6		0.2
MLD [Gy]	1.7 ± 1.4	-	0.5
D5 [Gy]	1.8 ± 1.6	<5.7	0.6
D2 [Gy]	2.1 ± 2.1	-	0.8
D1 [Gy]	0.6 ± 2.9	-	0
V5 [%]		-	
**Total lung**			
MLD [Gy]	2.5 ± 0.8	-	3.6 *
D2 [Gy]	18.2 ± 6.5	-	30.5 *
D1 [Gy]	25.0 ± 7.5	-	33.9 *
V5 [%]	15.4 ± 7.0	-	27.0 *
V10 [%]	5.3 ± 2.8	-	9.0 *
V20 [%]	1.6 ± 1.1	-	4.0 *
V30 [%]	0.5 ± 0.6	-	2.0 *

MLD: mean lung dose; Gy: Gray; SD: standard deviation. *: value > 97.5 percentile of the 170 patients included in the trial.

**Table 2 cancers-14-03520-t002:** Typical ipsilateral lung doses of radiotherapy of breast cancer related to different treatment scenarios.

**Reference**	**Data Source**	**Sample Size**	**Radiotherapy Scenario**	**Lung dose Parameters**
Current work	Prospective, clinical trial	170	External beam, 3D-CRT APBI (TD: 38 Gy, 10 fx)	V20: 3% MLD: 11%
Recht et al. [16]	Prospective, clinical trial	198	External beam, 3D-CRT APBI (photons, electrons, protons, TD: 32-36 Gy, 8–9 fx)	MLD: 1.4–4%
Stelczer et al. [15]	In silico study	40	External beam, 3D-CRT APBI (TD: 36.9 Gy, 9 fx)	MLD: 6%
		40	External beam, IMRT APBI (TD: 36.9 Gy, 9 fx)	MLD: 7–10%
Herein et al. [14]	In silico study	32	Multicatheter, interstitial HDR APBI (4× 6.25 Gy)	MLD: 5%
		32	Stereotactic, external beam APBI (4× 6.25 Gy)	MLD: 5%
Vasiljevic et al. [19]	Prospective, clinical trial	100	external beam WBI/chest wall irradiation (TD: 50–60 Gy, 25–30 fx)	V20: 8–13%
Jensen et al. [20]	In silico study	22	Free breathing, VMAT WBI (TD: 50 Gy, 25 fx)	V20: 13% MLD: 12 %
			3D-CRT, DIBH WBI (TD: 50 Gy, 25 fx)	V20: 9% MLD: 15%
Thomsen et al. [21]	Prospective, clinical trial	917	3D-CRT WBI (TD: 40 Gy, 15fx)	V17: 14–17%
		937	3D-CRT WBI (TD: 50 Gy, 25fx)	V20: 14–17%
Oechsner et al. [22]	In silico study	31	Free breathing, 3D-CRT WBI (TD: 50 Gy, 25 fx)	V20: 19% MLD: 20%
			DIBH, 3D-CRT WBI (TD: 50 Gy, 25 fx)	V20: 14% MLD: 16%
Gaál et al. [23]	Prospective, clinical trial	54	Free breathing, 3D-CRT WBI (TD: 50 Gy, 25 fx)	V20: 12% MLD: 14%
			DIBH, 3D-CRT WBI (TD: 50 Gy, 25 fx)	V20: 11% MLD: 13%
Pandeli et al. [24]	In silico study	20	Free breathing, 3D-CRT WBI (TD: 40 Gy, 15 fx)	V20: 8% MLD: 11%
			DIBH, 3D-CRT WBI (TD: 40 Gy, 15 fx)	V20: 8% MLD: 11%

Gy: Gray; TD: total reference dose; fx: fraction; 3D-CRT: 3D-conformal radiotherapy; APBI: accelerated partial breast irradiation; V20Gy: lung volume receiving ≥ 20 Gy; MLD: mean ipsilateral lung dose in percent of the reference dose [%]; IMRT: intensity-modulated radiotherapy; HDR: high-dose-rate; WBI: whole breast irradiation; VMAT: volumetric arc therapy; DIBH: deep inspiration breath hold.

## Data Availability

The data presented in this study are available on request from the corresponding author. The data are not publicly available due to privacy restrictions and ethical issues.

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
