# Peer review of "External Beam Accelerated Partial Breast Irradiation in Early Breast Cancer and the Risk for Radiogenic Pneumonitis"

_cancers, 2022, doi:10.3390/cancers14143520_

Round 1
Reviewer 1 Report
In this report the authors describe their cohort of patients from a German phase 2 trial treating low risk breast cancer after breast conserving surgery using accerated partial breast irradiation. Also they analyze in detail the one case of clinical radiation induced pneumonitis.
It is well written, scientifically sound and of interest to the community at large.
Some small typos should be corrected such as 2x "usually" in one sentence beginning at line 68, "titanium" instead of "titan"in line 97, "publication date" in line 123
Furthermore I suggest the authors should check the validity of the "non-coplanar" description in line 104. If correct, it should also be mentioned in the dicussion that this technique may increase lung dose compared to more tangential, coplanar plans. In the 2017 publication (6) the method "non-coplanar" is also described, however the coresponding two figures (in two planes) seem to show a coplanar plan; From the images schown in Figures 1 and 2 in this current article (which are from the same patient, the one with the pneumonitis I assume from the description in Figure 2), it seems a more tangential beam arrangement would decrease lung dose without a marked increase in contralateral breast dose. (Possibly anohter image in coronal and/or sagittal plane would add more information). In summary, to minimize confusion, this planar description could be omitted in my humble opinion or at least corrected to"...using (non-)coplanar fields..." . Also in this context one should mention the fact that deep INspiration techniques would likely also reduce Mean Lung Dose(MLD) as opposed to the end-EXpiration technique used by this group.
Please check Table 2: MLD is given in "%" throughout instead of dose(Gy). Is this a typo? If not, then correct the description accordingly.
I look forward to the long term locoregional control results.
Author Response
We thank reviewer 1 very much for the kind statements and the suggestions for improvement.
Point by point response:
- The mentioned small typing errors in lines 68 and 97 were corrected (see changes tracking). The point “publication date" in line 123 remained unclear to us.
- Regarding the radiotherapy field orientation (non-coplanar vs. coplanar) we follow the advice of the reviewer and omitted the term and used “tangential” instead.
- In the discussion we already referred to the beneficial use of DIBH: “Furthermore, the data presented in Table 2 document certain advantages for deep inspiration breath hold (DIBH) techniques compared to the use of free breathing techniques in WBI. The advantages of using APBI techniques appear to be even more pronounced. To date, the question whether different APBI techniques are systematically superior to others regarding the doses to the lungs cannot be answered adequately because of the absence of comparative prospective data.”
Therefore, we didn’t add further explanations.
- Table 2: we checked the values again. It is right the way it is. Because of differences in reporting of MLD values in the literature, we recalculated the MLD as percentages of the reference dose. We clarified this in the table subtext.
Reviewer 2 Report
This paper, while not presenting surprising findings, is still an important contribution to the published literature on the merits of APBI, in this case related to pneumonitis risks.
Some more information on the follow up and how pneumonitis was assessed would contribute to this paper (patient questionnaires, imaging, imaging only if symptoms....what was the process?)
The CTCAE grading system calls radiographic only pneumonitis Grade 1, and symptomatic pneumonitis Grade 2. Thus I would consider the pt in this study to have Grade 2 acute pneumonitis.
The paragraph about drawing a blood sample on the patient with pneumonitis 5 years after therapy does not flow with the rest of the methods and materials, and seems almost an afterthought once they had their findings. Was this planned all along? If so then it should be re-worded as part of the plan upfront and moved to the end of the section.
I was curious to know why such a large PTV margin of 1.5cm was chosen given that daily imaging was utilized during treatments?
Also curious to know if the authors would recommend a V20<3% for other fractionation schedules and how dose per fraction might influence these parameters.
Author Response
We thank reviewer 2 very much for the kind statements and the suggestions for improvement.
Point by point response:
- Reviewer: “Some more information on the follow up and how pneumonitis was assessed would contribute to this paper (patient questionnaires, imaging, imaging only if symptoms....what was the process?)”
Response:
We added some details to the following sentences in the materials and method section:
Follow-up was scheduled every 3 months for 2 years following breast conserving surgery, every 6 months in the years 3-5, and at 12-months intervals thereafter. Local control and survival parameters, as well as toxicity and cosmetic outcomes were assessed with mammograms at national guideline-based regular intervals and clinical examinations. Further imaging, e.g. computed tomography scans, were added in case of symptoms.
- Reviewer: “The CTCAE grading system calls radiographic only pneumonitis Grade 1, and symptomatic pneumonitis Grade 2. Thus I would consider the pt in this study to have Grade 2 acute pneumonitis.
Response: This comment of the reviewer is right. We checked the CTCAE v3.0 table again and changed the grading of the pneumonitis to Grade 2 throughout the text. Thank you very much.
- Reviewer: The paragraph about drawing a blood sample on the patient with pneumonitis 5 years after therapy does not flow with the rest of the methods and materials, and seems almost an afterthought once they had their findings. Was this planned all along? If so then it should be re-worded as part of the plan upfront and moved to the end of the section.
Response: The reviewer is right again. It was not part of the studyplan because we didn’t expect any pneumonitis (RP) with APBI at all. In fact, the patient experienced a symptomatic RP without increased lung dose values. Therefore, we had the idea to look for an increased individual radiosensitivity – and we found it. In our eyes this is a very important finding that should be considered more when looking at early and late toxicity results – and it was discussed that way. Because it was not planned we decided not to change the order of the manuscript.
- Reviewer: I was curious to know why such a large PTV margin of 1.5cm was chosen given that daily imaging was utilized during treatments?
Response: When we wrote the study protocol initially, we planned to apply the radiotherapy with free breathing and to compensate for that motion we chose this margin from CTV to PTV. However, when we started the study we changed the procedure to that endexspiratory breath hold technique as described, because that was the method we could monitor the best at this time. Furthermore, we did not reduce the margins then because of the surgery technique established in Germany: in contrary to the colleagues in the USA and some European countries (e.g. Hungary) which use open cavity surgery, in Germany the wound cavities are surgically adapted before closing the skin to avoid seroma and achieve better cosmetic results, which is called closed-cavity breast conserving surgery. This is leading to some more uncertainties for target delineation compared to closed cavity surgery. Therefore we decided to stick with our previous concept throughout the whole trial. Moreover, volume comparisons for example with PTVeval volumes from our Hungarian APBI colleagues after open cavity surgery show similar values, which supports our decision to maintain the 1.5 cm expansion.
- Reviewer: Also curious to know if the authors would recommend a V20<3% for other fractionation schedules and how dose per fraction might influence these parameters.
Response: In the discussion, we referred to the data of two other publications dealing with RP after APBI. One used a comparable fractionation (8-9x 4 Gy) and found more comparable results, the other one used 5x 6 Gy/10d and found no significant influence for a ipsilateral MLD V20<3% constraint. Data is sparse, therefore we cannot say more on this topic.